# Preserving the Authenticity of ST25 Rice (*Oryza sativa*) from the Mekong Delta: A Multivariate Geographical Characterization Approach

Dinh Tri Bui [1], Ngoc Minh Truong [1], Viet Anh Le [1], Hoang Khanh Nguyen [1], Quang Minh Bui [1], Van Thinh Pham [2] and Quang Trung Nguyen [1,*]

[1] Center for Research and Technology Transfer, Vietnam Academy of Sciences and Technology (VAST), 18 Hoang Quoc Viet Street, Cau Giay District, Hanoi 100000, Vietnam; bdtri@vast.vn (D.T.B.); minhtn689@gmail.com (N.M.T.); vietanh.livelearn@gmail.com (V.A.L.); nhk261097@gmail.com (H.K.N.); bui_quang_minh@yahoo.com (Q.M.B.)

[2] Faculty of Food Science and Technology, Ho Chi Minh University of Food Industry, 140 Le Trong Tan Street, Tan Phu District, Ho Chi Minh City 700000, Vietnam; phamvanthinh27@gmail.com

\* Correspondence: nqtrung79@gmail.com; Tel.: +84-4-3756-8422 (ext. 100000)

**Abstract:** The research centers around ST25, a recently acclaimed rice variety lauded as Vietnam's premier offering. However, its ability to substantiate its origin is impeded by the rampant proliferation of counterfeit derivatives within the market. A distinctive methodology is posited herein, intertwining the attributes of Fourier Transform Infrared Spectroscopy (FTIR) and Inductively Coupled Plasma Mass Spectrometry (ICP-MS) analyses, augmented using Principal Component Analysis (PCA). The primary objective is to meticulously ascertain the unadulterated geographic provenance of the ST25 rice cultivar. The findings unequivocally underscore the emergence of a conspicuous taxonomy within the ST25 rice samples sourced from Soc Trang, underpinned by the utilization of both FTIR and ICP-MS datasets. Remarkably, the discernment of eight elemental constituents ($^{27}$Al, $^{59}$Co, $^{44}$Ca, $^{57}$Fe, $^{60}$Ni, $^{63}$Cu, $^{93}$Nb, and $^{98}$Mo) has been adjudicated as pivotal in ascribing geospatial classification. The ramifications of this proposed modality encompass not only the authentication of the subject rice variety but also extend to the validation of similar grain types. Functioning as a potent deterrent against the omnipresent specter of food counterfeiting within the market milieu, this methodology occupies a pivotal niche.

**Keywords:** *Oryza sativa*; ST25; ICP-MS; FTIR; geographical classification; PCA

## 1. Introduction

Vietnam is situated among the foremost quintet of countries, alongside India, Thailand, Pakistan, and China, in global rice production and export [1]. During the year 2020, Vietnam attained second position as the largest rice exporter worldwide, dispatching a cumulative volume of 6.15 million tons of rice [2], where 80% comprised premium-grade, high-value rice varieties [3]. Recent times have witnessed the ascendancy of numerous Vietnamese rice cultivars, notably including the ST25 strain, onto the international stage, thereby clinching eminent positions within the prestigious World's Best Rice Contest [3]. The ST25 variety, an innovation credited to the agricultural luminary Ho Quang Cua, hailing from the province of Soc Trang, has secured the accolade of first and second prizes in the 2019 and 2020 iterations of the contest, respectively, thus propelling its prominence to unprecedented heights both within domestic and global realms [4]. Within the domestic market, ST25 occupies the echelons of a premium rice segment, commanding prices nearly 2.5 times higher than those associated with medium-quality rice [5]. On the international front, ST25 has not only carved a niche for itself but also demonstrated a formidable presence in markets spanning the United States, Australia, and Japan, underpinned by its safeguarded trademarks and adherence to exacting quality standards [6–8].

Despite the burgeoning acclaim of ST25 rice, the specter of food fraud looms as a global apprehension, casting a shadow over consumer health and well-being. An array of investigations has illuminated the pervasive adulteration of comestibles, including aromatic spices and botanical herbs, primarily motivated by economic incentives [9]. Notwithstanding the absence of documented instances of ST25 rice adulteration within Vietnam, a disconcerting prevalence of alternative rice cultivars parading as "ST25 rice" has emerged, capitalizing on resemblances in morphology and intrinsic attributes [10]. Moreover, several provinces have undertaken the importation of ST25 rice seeds from the epicenter of its cultivation, Soc Trang, with the intention of engendering their own yield of ST25 rice products. This phenomenon poses a quandary in differentiating authentic ST25 rice from spurious counterparts, barring the discernment of erudite experts seasoned in rice provenance identification [5]. Consequently, customers find themselves entangled in a quagmire of uncertainty when attempting to gauge the authenticity of ST25 rice merchandise.

To counteract the burgeoning malaise of food counterfeiting, the adoption of authentication strategies grounded in quantitative assessment has proven efficacious in elucidating the origin of alimentary commodities [11]. The synergy of chemometric analysis and empirical data gleaned from analytical methodologies, such as Fourier-transform infrared (FTIR) spectroscopy, have emerged as a dependable panacea [12]. FTIR, distinguished by its expeditiousness, cost-effectiveness, and non-destructive attributes, has, in concert with chemometrics, engendered success in discerning dissimilarity among diverse food specimens, encompassing oils and spices [12]. In parallel, elemental profiling facilitated using Inductively Coupled Plasma Mass Spectrometry (ICP-MS), harmonized with chemometric techniques, has unveiled its potential in ascertaining the geographical origin of victuals [13]. Illustratively, this amalgamation has enabled the differentiation of wines derived from disparate terroirs and the classification of cabbage cultivars predicated on distinctive inorganic compositional footprints [14–17]. A staple of such analysis, Principal Component Analysis (PCA), constitutes a conventional approach to distill valuable insights from ICP-MS data, effectuating the visualization of variances within samples and unearthing elemental moieties pivotal in classification endeavors [13].

In the present inquiry, we unfurl an innovative and potent paradigm by synergistically employing FTIR and ICP-MS methodologies in conjunction with PCA to unveil the geospatial provenance of the ST25 rice cultivar. The crux of our endeavor resides in bestowing a dependable recourse for the authentication of this venerable rice strain alongside its analogs in the rice domain. Using meticulous inquiry, our primary mission gravitates toward the resolution of the impending conundrum surrounding the verification of ST25 rice provenance and the effective counteraction of the mushrooming proliferation of counterfeit commodities within the market milieu. By harnessing the idiosyncratic spectral and elemental attributes intrinsic to ST25 rice from disparate locales, our pioneering methodological schema furnishes a promising tool to undergird the inviolability and verity of this extraordinary rice cultivar, thereby nurturing consumer trust and fortifying the bedrock of food security.

## 2. Results and Discussion

### 2.1. Organoleptic Characteristics of ST25 Rice Samples

The ST25 grain is described as approximately 7–9 mm in length, thin, white, and clear, lacking any silver coloration [18]. Once cooked, it exudes a unique aroma reminiscent of pandan (*Pandanus amaryllifolius*) leaves and immature glutinous rice (*Oryza sativa* var. *glutinosa*) kernels [19]. Notably, its texture retains a pleasing softness even as it cools [18]. Figure 1 presents the morphology and distinguishing features of the ST25 variety from three distinct areas. Upon evaluating its organoleptic characteristics, little to no differences were observed among the samples from these regions. As a result, relying solely on organoleptic traits for classifying the origins of ST25 rice proved to be impractical.

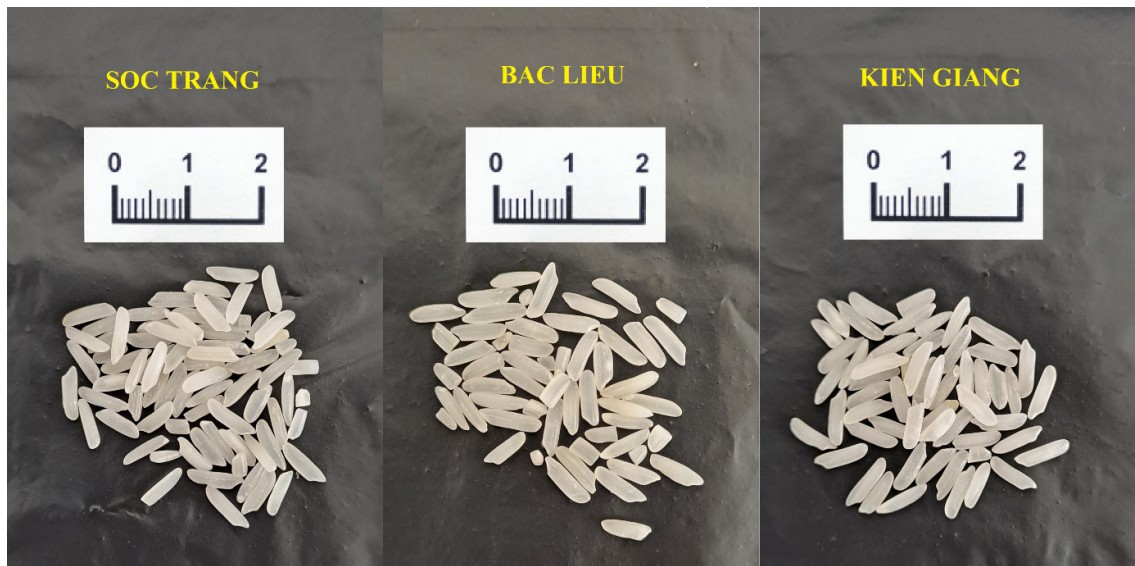

**Figure 1.** Organoleptic characteristics of ST25 rice grown in 3 different areas.

*2.2. Infrared Spectroscopy Analysis*

Figure 2 depicts the comprehensive infrared spectra of ST25 rice samples originating from distinct regions within the Mekong Delta. Evidently, specific peaks that bear the hallmarks of unique rice sample features were meticulously chosen for analysis. These infrared spectra furnish profound insights into the vibration frequencies attributed to specific functional groups, which correspond to compound classes previously identified and documented [20] (Table 1). Notably, a distinctive band residing at 3294 $cm^{-1}$, positioned within the spectral range of 3600–3000 $cm^{-1}$, was attributed to the stretching vibrations of hydroxyl (−OH) and amino (−NH) groups. Moreover, the presence of stretching vibrations pertaining to carbon-hydrogen (−CH) bonds at 2928 $cm^{-1}$, situated within the 3000–2500 $cm^{-1}$ region, indicated the potential existence of alkene, alkyne, and alkane compounds [21,22]. Evident sharp peaks manifesting at wavelengths of 1638, 1148, and 1077 $cm^{-1}$ correspond to the stretching of carbonyl (–C=O), carbon–carbon double bond (–C=C), and carbon–oxygen (–C–O) bonds, respectively. These vibrational phenomena are indicative of compound classes such as alkenes and alcohols. Notably, the vibrational range spanning from 995 to 413 $cm^{-1}$, commonly referred to as the "Fingerprint region", played a pivotal role in characterizing the principal attributes of the ST25 rice samples [21]. Despite the wealth of spectral data, the complete spectra in isolation were found insufficient to effectively differentiate between the ST25 rice samples originating from the distinct regions. This inadequacy stemmed from the marginal variations present within the spectral signals across the three regions. As a remedy to achieve successful differentiation and classification, we employed multivariate statistical analysis (MSA), with a particular emphasis on Principle Component Analysis (PCA).

**Table 1.** Corresponding functional groups and possible compound classes of FTIR signals of ST25 rice sample.

| Wavelength Region (cm$^{-1}$) | Selected Peaks | Vibration of Functional Groups | Primary Compound Class | References |
|---|---|---|---|---|
| 3600–3000 | 3294 | O–H and N–H groups (Stretching vibration) | Polysaccharides, amine, alcohol | [21,22] |
| 3000–2500 | 2928 | C–H stretching | Alkyne, alkene, alkane | [21,22] |
| 1670–1600 | 1638 | C=O and C=C stretching | Alkene | [21,22] |

**Table 1.** *Cont.*

| Wavelength Region (cm$^{-1}$) | Selected Peaks | Vibration of Functional Groups | Primary Compound Class | References |
|---|---|---|---|---|
| 1600–1300 | 1540 | N-O stretching | Nitro compound | [21] |
| | 1413 | Rocking vibrations of CH bond | Cis-disubstituted alkenes | [21] |
| | 1337 | CH$_3$ bending vibrations | Lipids, proteins | [21,22] |
| 1300–1000 | 1236 | C–N stretching | Amine | [22] |
| | 1148 | C–O stretching | Alcohol, ester | [22] |
| | 1077 | C–O stretching | Alcohol | [22] |
| 1000–400 | 995–413 | "Fingerprint region" | - | [22] |

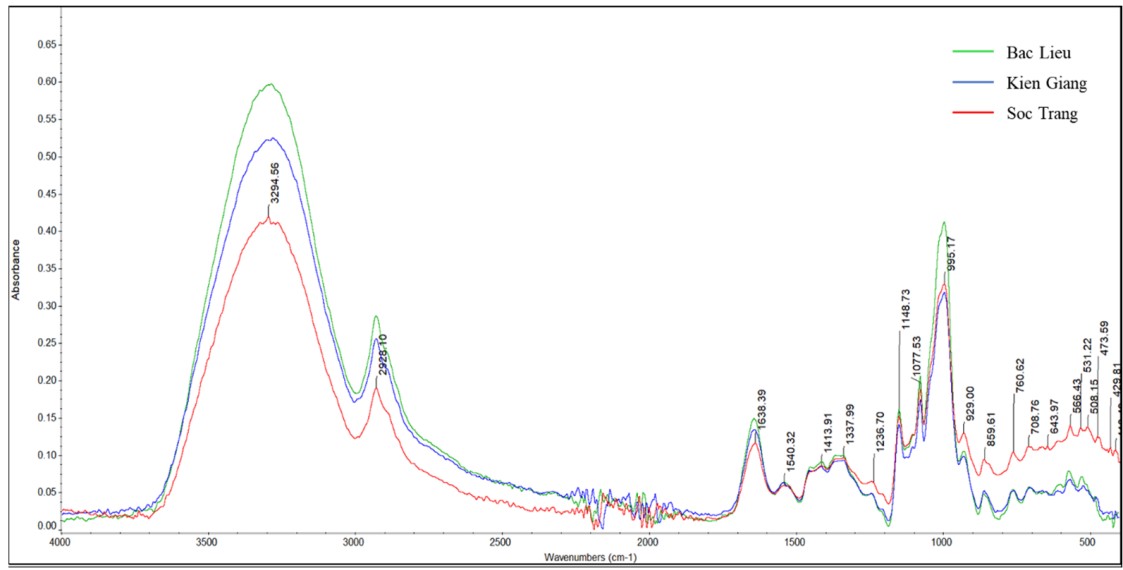

**Figure 2.** FTIR spectra of ST rice samples and selected peaks for data analysis.

### 2.3. Elemental Measurement Using ICP-MS

Table 2 displays the elemental composition of ST25 rice samples gathered from three distinct regions within the Mekong Delta: Soc Trang, Kien Giang, and Bac Lieu. The elemental analysis conducted unveiled the noteworthy abundance of $^{39}$K in ST25 rice samples, with concentrations spanning from 53.7 to 63.1 mg/g DW across the triad of regions. Following closely, the element $^{44}$Ca emerged as the second most prevalent, manifesting concentrations ranging from 14.5 to 45.0 mg/g DW across the three sample sets. Of particular significance is the discernible disparity in element levels among the Soc Trang, Bac Lieu, and Kien Giang samples. It is pertinent to note that ST25 rice from Bac Lieu exclusively featured all 24 elements, inclusive of a detectable presence of $^{93}$Nb at 0.06 mg/g DW. Conversely, the ST25 samples from Kien Giang exhibited an absence of $^{59}$Co. Evidently, the elemental constitution within the ST25 rice samples demonstrated substantial variations that can be attributed to the inherent geographical dissimilarities across the regions.

**Table 2.** Elemental contents of ST25 rice samples from 3 areas of the Mekong Delta.

| Element | Geographical Regions (mg/g DW) | | |
|---|---|---|---|
| | Soc Trang | Kien Giang | Bac Lieu |
| $^7$Li | 0.14 ± 0.01 | 0.13 ± 0.01 | 0.06 ± 0.002 |
| $^{23}$Na | 8.82 ± 0.33 | 6.78 ± 0.40 | 3.47 ± 0.15 |
| $^{24}$Mg | 16.62 ± 0.91 | 13.67 ± 0.71 | 15.11 ± 0.92 |

**Table 2.** *Cont.*

| Element | Geographical Regions (mg/g DW) | | |
|---|---|---|---|
| | Soc Trang | Kien Giang | Bac Lieu |
| $^{27}$Al | 1.10 ± 0.04 | 1.60 ± 0.07 | 0.57 ± 0.01 |
| $^{39}$K | 63.14 ± 3.18 | 53.72 ± 2.25 | 54.96 ± 3.45 |
| $^{44}$Ca | 45.07 ± 2.51 | 14.51 ± 0.58 | 16.81 ± 0.79 |
| $^{48}$Ti | 0.24 ± 0.02 | 0.19 ± 0.01 | 0.20 ± 0.01 |
| $^{51}$V | 0.05 ± 0.002 | 0.04 ± 0.001 | 0.03 ± 0.001 |
| $^{52}$Cr | 4.42 ± 0.33 | 1.59 ± 0.07 | 1.13 ± 0.05 |
| $^{55}$Mn | 1.67 ± 0.07 | 0.79 ± 0.05 | 1.34 ± 0.08 |
| $^{57}$Fe | 25.1 ± 1.32 | 7.43 ± 0.56 | 9.57 ± 0.46 |
| $^{59}$Co | 0.04 ± 0.002 | - | 0.02 ± 0.001 |
| $^{60}$Ni | 2.71 ± 0.12 | 0.57 ± 0.04 | 0.80 ± 0.06 |
| $^{63}$Cu | 0.31 ± 0.02 | 0.06 ± 0.003 | 0.21 ± 0.01 |
| $^{66}$Zn | 6.66 ± 0.35 | 3.70 ± 0.22 | 5.05 ± 0.24 |
| $^{93}$Nb | - | - | 0.06 ± 0.003 |
| $^{98}$Mo | 0.07 ± 0.003 | 0.04 ± 0.002 | 0.08 ± 0.003 |
| $^{111}$Cd | 0.03 ± 0.001 | 0.03 ± 0.001 | 0.03 ± 0.001 |
| $^{115}$In | 0.03 ± 0.002 | 0.03 ± 0.002 | 0.03 ± 0.002 |
| $^{121}$Sb | 0.01 ± 0.00 | 0.01 ± 0.00 | 0.01 ± 0.00 |
| $^{138}$Ba | 0.65 ± 0.02 | 0.82 ± 0.03 | 0.76 ± 0.03 |
| $^{202}$Hg | 0.03 ± 0.001 | 0.02 ± 0.001 | 0.02 ± 0.001 |
| $^{208}$Pb | 0.11 ± 0.004 | 0.11 ± 0.01 | 0.12 ± 0.01 |
| $^{209}$Bi | 0.09 ± 0.01 | 0.09 ± 0.004 | 0.09 ± 0.004 |

*2.4. Geographical Classification Using Principle Component Analysis*

2.4.1. Geographical Classification Based on FTIR Dataset

To discern the intricate interplay of spectral signals and unravel the influence of geographical origins—Soc Trang, Bac Lieu, and Kien Giang—on the chemical composition of ST25 rice samples, we leveraged the robust analytical tool of Principal Component Analysis (PCA). In a meticulous manner, we handpicked specific peaks at distinct wavelengths, as meticulously documented in Table 1, designating them as variables to underpin the statistical procedures. Subsequently, we marshaled the corresponding absorbance values, meticulously derived from the intricate spectra, to construct a novel data matrix poised for the forthcoming PCA interrogation. This intricate data matrix underwent rigorous PCA processing, meticulously executed via the employment of XLSTAT 2016.02.28451 (Addinsoft, Paris, France).

The underlying contribution of the Principal Components (PCs) in explicating the novel plot's intricacies finds a graphical portrayal in Figure 3a, in which the blue bars represents the eigenvalue of each PCs and the red-dotted line indicates the cumulative variability of the PCs. A notably salient observation emerges: PC1 and PC2 jointly accounted for a staggering excess of 99% of the cumulative variance inherent in the entire sample cohort. This observation succinctly underscores that these two pivotal Principal Components efficaciously encapsulate the lion's share of the information enveloped within the FTIR matrix dataset. Notably, this prodigious accumulation of variance was accompanied by a crucial phase of projecting and interpreting the FTIR dataset afresh, albeit within the newfound coordinate system predicated upon the orthogonal PCs, PC1 and PC2. This transformative exercise yielded a Score plot, splendidly etched in Figure 3b, which effectively elucidated the positioning of ST25 rice samples originating from Soc Trang. The discerning eye, upon perusing this visualization, would readily discern the emergence of distinct clusters, unmistakably demarcating these samples from their counterparts drawn from the other two geographical domains. However, in the same breath, it is imperative to acknowledge that a modicum of perplexity emerged in the disposition of ST25 rice samples sourced from Kien Giang and Bac Lieu and, to a lesser extent, even from Soc

Trang. Regrettably, these enigmatic samples defied clear demarcation, eluding successful clustering and classification endeavors.

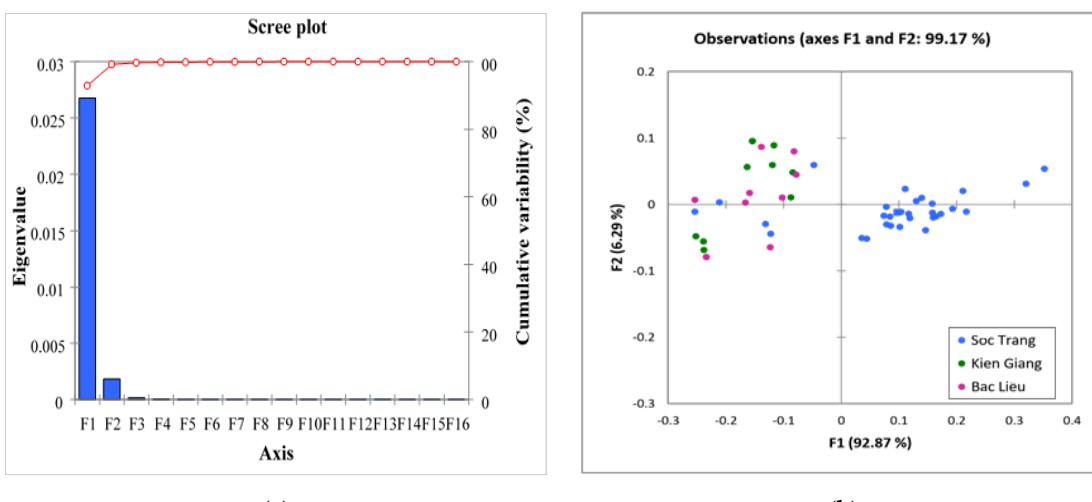

(**a**)                                                                    (**b**)

**Figure 3.** PCA Scree plot (**a**) and score plot (**b**) of FTIR dataset.

2.4.2. Geographical Classification Based on ICP-MS Dataset

In contrast to the methodology applied in FTIR analysis, where preprocessing of the dataset was conducted prior to Principle Component Analysis (PCA), the elemental composition of the ST25 rice samples was employed directly as the dataset for algorithmic processing. This entailed the consideration of content variations in the 24 elemental components across samples originating from diverse geographical regions as independent variables. The utilization of Inductively Coupled Plasma Mass Spectrometry (ICP-MS) data was characterized by a distinct analytical approach, yielding results that exhibited greater separability compared to the outcomes of FTIR analysis.

Using the application of PCA, the ICP-MS dataset was translated into a two-dimensional space represented using the first two Principal Components (PCs), as illustrated in Figure 4. This visual representation effectively displayed a more pronounced distinction between the ST25 rice samples, classifying them according to their geographical origins: Soc Trang, Bac Lieu, and Kien Giang. Approximately 47% of the total information encompassed by the dataset was encapsulated by the score scatterplot, with PC1 and PC2 contributing 31.23% and 15.90%, respectively. The fundamental insights gleaned from this analysis are rooted in the arrangement of samples within this transformed space.

Further insight into the factors contributing to the classification of geographical origin was garnered using the loading scatterplot depicted in Figure 5. This scatterplot effectively conveyed the correlation between each elemental component (variable) and its influence on the classification of the samples' geographical origins. Notably, elements $^{57}$Fe, $^{60}$Ni, and $^{44}$Ca displayed robust correlations, exerting significant influence on the separation observed along PC1, with correlations nearing unity. Conversely, elements $^{27}$Al and $^{93}$Nb emerged as pivotal factors impacting the classification along PC2. It's worth noting that due to their divergent distribution patterns within the plot, these two elements exhibited a negative correlation, highlighting their contrasting contribution to the classification process.

The elemental composition of ST25 rice samples underwent rigorous analysis, and the outcomes were succinctly portrayed using moving range charts (Figure 6). It is notable that among these elemental constituents, namely $^{44}$Ca, $^{57}$Fe, $^{59}$Co, $^{60}$Ni, and $^{63}$Cu, a conspicuous augmentation in concentration was observed within ST25 rice samples originating from Soc Trang in contrast to those harvested in Kien Giang and Bac Lieu (Figure 6b–f). This collection of elements, in essence, serves as distinct markers, unequivocally delineating the distinctive identity of Soc Trang's ST25 rice.

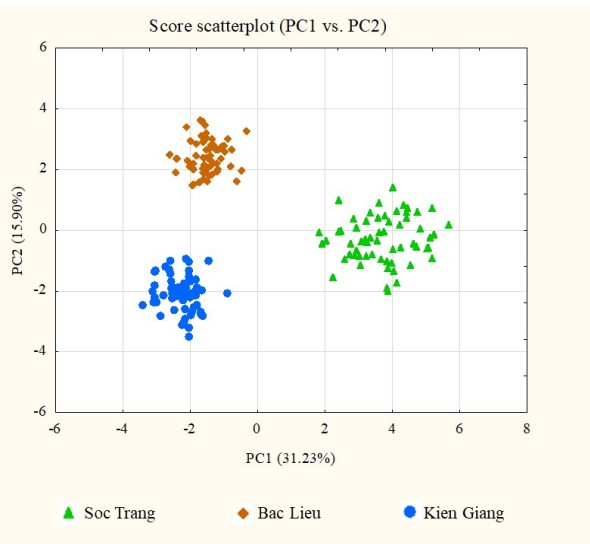

**Figure 4.** PCA score scatterplot of the ICP-MS dataset.

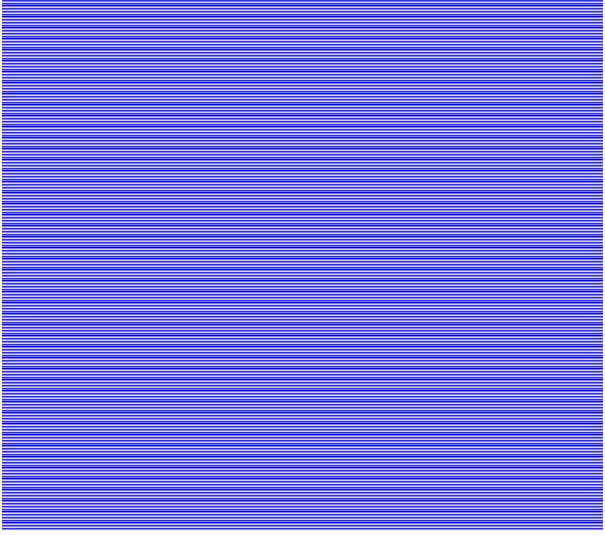

**Figure 5.** Loading scatter plot of 24 elements.

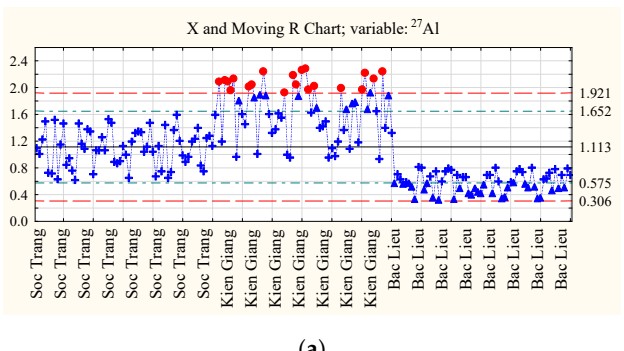

(**a**)

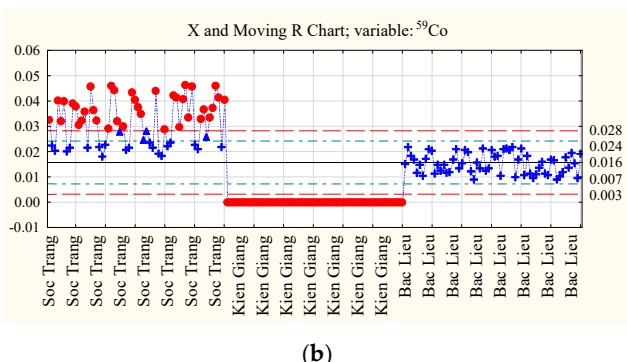

(**b**)

**Figure 6.** *Cont*.

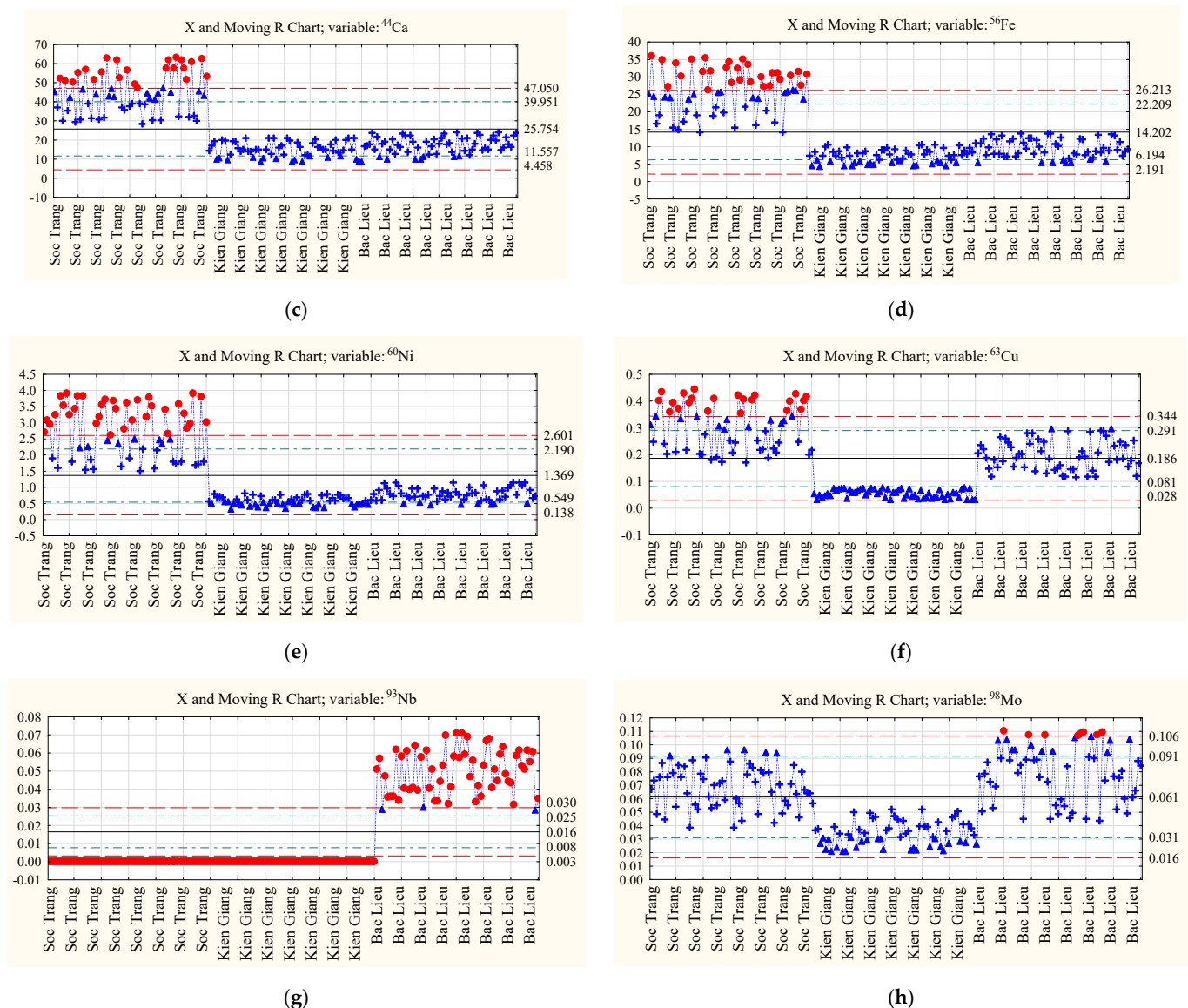

**Figure 6.** Moving range chart of the discriminating elements in ST25 rice samples from different areas. (**a**) $^{27}$Al; (**b**) $^{59}$Co; (**c**) $^{44}$Ca; (**d**) $^{57}$Fe; (**e**) $^{60}$Ni; (**f**) $^{63}$Cu; (**g**) $^{93}$Nb; and (**h**) $^{98}$Mo. The red color indicates high concentration levels. The different shapes of blue elements indicate various elements, and the blue color signifies relatively low concentrations. Red Dots: The red dots in Figure 6 represent data points corresponding to specific rice samples, and their positioning on the graph signifies the concentration levels of certain isotopes or attributes. In this context, red dots with higher positions on the graph indicate elevated concentrations of the mentioned attributes, while those with lower positions suggest subdued or even zero concentrations. Different Shape Blue Elements: The different shapes of blue elements on the graph likely represent distinct rice samples or groups of samples. Each shape corresponds to a particular geographical origin or some other grouping criterion. For instance, if circles represent rice samples from Kien Giang, triangles may represent those from Bac Lieu, and squares could be used for Soc Trang samples. The specific shapes and their associated colors are used to differentiate between the various sample groups.

Concurrently, it was ascertained that isotopes $^{27}$Al and $^{59}$Co played pivotal roles as discerning factors for rice samples procured from Kien Giang. Notably, their concentrations were discernibly elevated and subdued (attaining equivalence to zero) relative to other samples, respectively (Figure 6a,b). Intriguingly, the two discriminative attributes for Kien Giang rice samples exhibited a discernible negative correlation. Contrastingly, Bac Lieu rice

samples exhibited a conspicuous prevalence of element [93]Nb, exclusively segregating them, while the content of [98]Mo in these samples experienced a considerable spike, surpassing that in Soc Trang and Kien Giang samples. These particular elements, [93]Nb and [98]Mo, unambiguously emerged as the principal discriminators characterizing Bac Lieu's rice samples (Figure 6g,h). The prominence of these elemental differentials imparts invaluable insights into the stratification of ST25 rice samples originating from diverse geographical domains, thereby potentially furnishing a potent avenue for the substantiation of authenticity.

## 3. Materials and Methods

### 3.1. Sample Collection and Preparation

A meticulous collection of a total of 90 samples of ST25 rice was undertaken, hailing from three discrete provinces—Soc Trang, Bac Lieu, and Kien Giang—located in close adjacency within the Mekong Delta region of Southeast Vietnam, as illustrated in Figure 7. In preparation for subsequent analytical investigations, the procured rice grains were subjected to a grinding process utilizing a Seka 800Y grinder sourced from Japan, followed by sieving to attain particle dimensions ranging from 0.125 to 0.150 mm. In anticipation of the analytical phase, a preparatory regimen was administered to the samples. This involved subjecting the collected rice to a pre-treatment procedure, encompassing a desiccation stage executed within a Memmert UN110 drying oven. This drying process was conducted at a consistent temperature of 80 °C over a temporal span of 3 h, thereby effecting the eradication of any lingering residual moisture. Subsequent to desiccation, the resultant dried samples were securely ensconced within hermetically sealed zipper bags, which were, in turn, stored within a controlled environment comprising a silica gel chamber. This method of storage was meticulously adopted to ensure the sustained preservation of the sample's structural integrity until the forthcoming investigative phase.

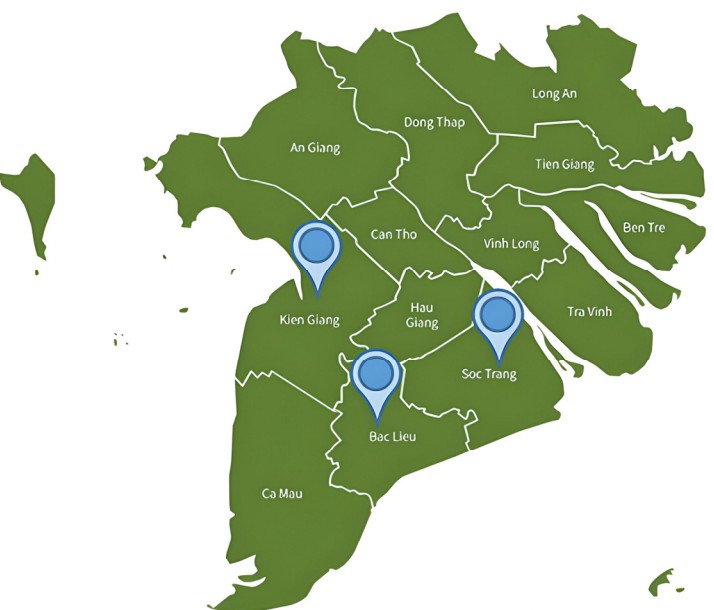

**Figure 7.** Geographical areas for ST25 rice samples collection.

### 3.2. Chemicals and Reagents

Concentrated nitric acid ($HNO_3$–65%) and hydrogen peroxide ($H_2O_2$–30%) solutions were acquired from Merck, Readington, NJ, USA, while deionized water (18.2 M$\Omega$cm) was generated using a Milli-Q Plus system (Millipore, Burlington, MA, USA). Standard solutions containing twenty-six multi-elements (B, Na, Mg, Al, Si, K, Ca, V, Cr, Mn, Fe, Co, Ni, Cu, Zn, Ga, As, Rb, Sr, Ag, Cd, Sb, Cs, Ba, Hg, and Pb) (TraceCERT) and nine rare earth elements (Sc, Y, La, Ce, Pr, Nd, Sm, Eu, and Gd) (10 mg/L of each) were sourced from Sigma-Aldrich Company (St. Louis, MO, USA). Sensitivity factors for all elements

in the diluted samples for the semi-quantitative mode were determined using a standard solution of Ti, Zr, Nb, Mo, Pd, Sn, and W (50 μg/L of each) in 6% ethanol and 0.14 M $HNO_3$. Absolute methanol was employed for matrix-matched standards preparation, but it was omitted in the calibration standard for measuring the digested sample.

### 3.3. FTIR Analysis

Fourier Transform Infrared Spectrometry (FTIR) using a Nicolet™ iS50 instrument from Thermo Scientific, Waltham, MA, USA, was employed to gather the absorbance spectra of the rice samples. For the spectral data acquisition, two common methods were utilized with the dried samples: (1) pressing the samples into KBr pellets and (2) directly measuring them using the Attenuated Total Reflection (ATR) technique, which involves pressing the samples onto a high-refractive-index prism (ZnSe crystal). Considering the higher signal difference [22] and greater feasibility, the ATR method was chosen for spectral data collection in this study. To ensure measurement stability across replicates, a generous amount of sample powder was loaded onto the ZnSe crystal. The absorbance spectra were recorded within the 400–4000 $cm^{-1}$ range with 128 scans and a resolution of 8 $cm^{-1}$. Prior to each sample measurement, background subtraction was automatically performed. Each sample underwent quintuple measurements ($n$ = 5) to enhance the accuracy and reliability of the results.

### 3.4. Elemental Quantification

A rice powder sample of 0.2 g was directly weighed into PTFE vessels, followed by the careful addition of 4 mL of $HNO_3$ and 1 mL of $H_2O_2$ to the vessels. The vessels were left to stand and opened in a fume hood for 2 h before undergoing microwave digestion using the Mars 6 instrument (CEM, Matthews, NC, USA). The microwave operation program consisted of three stages: (i) raising the temperature to 90 °C within 10 min, holding for 5 min, then (ii) increasing to 130 °C within 10 min, holding for 5 min, and finally (iii) raising to 150 °C within 5 min, and holding for the last 10 min. After the mixture cooled to room temperature, the solution was diluted using ultrapure water in a 50 mL volumetric flask.

The elemental measurement was conducted using an Agilent 7900 ICP-MS instrument (Agilent Technologies, Tokyo, Japan). The elemental measurement conditions were set at 1550 W of RF power, 2.0 V of RF matching, −40 V, −60 V, and 5.0 V of cell entrance, exit, and energy, respectively, and 2 °C for the temperature of the spray chamber. Argon and helium were used as the carrier and auxiliary gases, with flow rates set at 1.09 L/min and 4.3 L/min, respectively. Elemental quantification results were obtained using calibration curves constructed from multi-element standards prepared in 1% $HNO_3$.

### 3.5. Data Treatment and Statistical Analysis

FTIR spectral data were transformed into a spreadsheet using Unscrambler® X (Version 10.4.43636.111, CAMO Software, Oslo, Norway). STATISTICA (Version 12.5.192.0, Stat Soft. Inc., St. Tulsa, OK, USA) and XLSTAT (Version 2016.02.28451, Addinsoft, Paris, France) were employed to perform the algorithm of Principle Component Analysis (PCA).

## 4. Conclusions

In the present study, we embarked on an innovative exploration into the realm of geographical classification, a pursuit that entailed the amalgamation of FTIR and ICP-MS analysis with the formidable PCA algorithm. Our endeavor was directed at unraveling the intricate tapestry of ST25 rice samples' origins. The tapestry that emerged from our laborious analysis unveiled discrete enclaves within the realm of ST25 rice, specifically those cultivated in Soc Trang, Bac Lieu, and Kien Giang. Notably, the PCA treatment of the ICP-MS dataset, while explicating only approximately 60% of its intricate dimensions, exhibited a profound superiority in statistical analytical performance when juxtaposed against FTIR in terms of its capacity to classify with enhanced accuracy.

Furthermore, our endeavor unveiled a compelling revelation—specific elements, the elemental avatars of $^{27}$Al, $^{59}$Co, $^{44}$Ca, $^{57}$Fe, $^{60}$Ni, $^{63}$Cu, $^{93}$Nb, and $^{98}$Mo—stepped forth as the linchpins of the intricate dance of ST25 rice classification, emblematic of their respective geographic origins. It became strikingly evident that these elemental harmonies exerted a more resonant influence in differentiating ST25 rice samples derived from disparate corners of the geographical spectrum, eclipsing even the spectral symphony of FTIR signals.

Indeed, the ST25 rice variety has solidified its supremacy as the discerning palate's premier preference, commanding an unwavering allegiance from Vietnamese connoisseurs and achieving eminence in the global gastronomic tableau. However, the shadow cast by the specter of counterfeit products looms ominously, a concern that reverberates both within the confines of Vietnam and resonates on the international stage. It is against this backdrop that our study assumes a poignant relevance as we proffer a ray of hope in the form of an authentication methodology. This methodology, a sentinel guarding the ramparts of authenticity, has the potential to breach the citadel of counterfeit and reclaim the integrity of ST25 rice's origin. Beyond this, our methodology may serve as a guiding star, illuminating the path to authenticity for not only ST25 rice but also for other rice varietals ensnared by the clutches of counterfeiting.

The pivotal role of identifying bona fide ST25 rice amidst the labyrinthine maze of geographical sources is one that we have endeavored to underscore. The ramifications are profound, extending to instilling unshakable consumer confidence and fostering a sanctuary of trust within the intricate ecosystem of global food commerce. The resilience of ST25 rice's reputation stands fortified by the bastions of our method, guarding against the subterfuge of counterfeit incursions while also standing as a testament to our commitment to nurturing the livelihoods of farmers and producers who stake their endeavors on the authenticity of their produce.

Moreover, the profound significance of our proposed approach extends beyond the immediate sphere of ST25 rice authentication, resonating harmoniously with the broader symphony of combatting food fraud. By cultivating a culture of rigorous authenticity validation, our approach contributes to the broader endeavor of fortifying food security and engendering a sense of reliability in the intricate global network of sustenance.

**Author Contributions:** Conceptualization, D.T.B. and N.M.T.; methodology, V.A.L. and H.K.N.; software, V.A.L., N.M.T. and V.T.P.; validation, D.T.B. and Q.M.B.; formal analysis, N.M.T. and D.T.B.; investigation, H.K.N.; resources, V.T.P. and Q.T.N.; data curation, N.M.T.; writing—original draft preparation, V.A.L.; writing—review and editing, V.T.P., N.M.T. and Q.T.N.; visualization, Q.M.B. and H.K.N.; supervision, D.T.B. and Q.T.N.; project administration, V.T.P. and H.K.N.; funding acquisition, D.T.B. All authors have read and agreed to the published version of the manuscript.

**Funding:** This research was funded by the Vietnam Academy of Science and Technology (VAST) under grant number NCVCC02/22-23.

**Data Availability Statement:** The data presented in this study are available on request from the corresponding author.

**Acknowledgments:** Several researchers from the Center for Research and Technology Transfer, Vietnam is, appreciated for supporting and giving helpful advice for our study.

**Conflicts of Interest:** The authors declare no conflict of interest.

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
