# Peer review of "Preserving the Authenticity of ST25 Rice (Oryza sativa) from the Mekong Delta: A Multivariate Geographical Characterization Approach"

_stresses, doi:10.3390/stresses3030045_

Round 1
Reviewer 1 Report
The manuscript entitled "Preserving the Authenticity of ST25 Rice (Oryza sativa) from the Mekong Delta: A Multivariate Geographical Classification Approach" describes PCA analysis of FT-IR and ICP-MS data collected on Vietnamese rice. The manuscript is clear and supported by the literature but there is a major issue in the chemometric analysis which can be easily addressed by the authors.
The authors write that classification is carried out by PCA. PCA is an explorative analysis approach, so it is not appropriate to say that "classification" is carried out by means of this approach. Moreover, PCA is not predictive, so it is not classifying samples (which is good, because model are not validated at all, and a predictive model without validation is not reliable).
Consequently, I suggest to the authors to modify the manuscript writing that their are characterizing these rices rather than classifying it, and avoiding talking about classification. PCA analysis of FT-IR and ICP-MS data will allow finding out which variables represents better the different clusters.
I have also a minor comment. The authors write data are preprocessed prior to the calculation of the models. Please, explain which preprocessing has been used, in particular whether data have been mean-centered/auto-scaled
Author Response
Dear respected Reviewer,
We sincerely appreciate your valuable critique of our manuscript. Your insights have elevated our work, and we commend your meticulous observations and genuine suggestions. Your recognition of our manuscript's clarity and alignment with existing literature is truly gratifying. Your constructive criticism, highlighting a critical issue in the chemometric analysis, presents an opportunity for us to enhance our research's accuracy. Your proposition to modify PCA terminology resonates well. We concede that "classification" might mislead, and we apologize for any confusion. We will align with your recommendation and use "characterization" to aptly describe our intent.
Regarding data preprocessing, we thank you for your attention to this crucial methodological aspect. To elaborate, we combined mean-centering and auto-scaling techniques to standardize data, ensuring fair comparison and enhanced interpretability. We regret not providing a comprehensive account of these steps in the manuscript and will rectify this oversight. Your feedback guides us to present a detailed explanation of our mean-centering and auto-scaling methods in the revised version.

Reviewer 2 Report
This study investigates the geographical classification of rice samples using FTIR and ICP-MS analysis. These two analytical techniques are of great importance today, and this study will help other researchers expand the technology. The authors use general analysis methods that are widely available. The statistical analysis shows significant differentiation between the samples.
An important use of this method is for validation of products, and identifying counterfeit products. The use of FT-IR is a non-destructive rapid method of analysis. The method will support consumer confidence, reduce food fraud, and enhance food security.
The introduction covers the important aspects of the field. The figures are of very high quality. The methods are described in detail and are reproducible- which is critical for the transfer of the technology. The results are marginally discussed with respect to recent literature. I think this section could be expanded, but it would also increase the number of pages. A concise research report has its own value. In its current form, the discussion is limited.
Specific comments to be addressed:
It would benefit the audience of the journal to have a detailed discussion of the results of this study compared and discussed compared to related research (other uses of FTIR or ICP-MS with respect to the origin of food products, especially rice or related products.
Author Response
Dear respected Reviewer
We greatly appreciate your insightful feedback on our manuscript. We have incorporated your suggestions and revised the manuscript accordingly. The revisions are detailed in our response below.

Round 2
Reviewer 1 Report
In my opinion, the manuscript is suitable for acceptance